# Analysis of related factors for adolescents' intention to use alcohol in Korea

**Eun-A Park**[1], **Ae-Ri Jung**[2]*, **Sungyong Choi**[3]

**1** Department of Nursing, Bucheon University, Bucheon-si, Gyeonggi-do, South Korea, **2** College of Nursing, Eulji University, Uijeongbu-si, Gyeonggi-do, South Korea, **3** Official, Big Data Division, Seoul Metropolitan Government, Jung-gu, Seoul, South Korea

These authors contributed equally to this work.
* aeri@eulji.ac.kr

**Data Availability Statement:** These data are third party data which are not owned or collected by the author. The data underlying the results presented in the study are available from Korea Health Promotion Institute. This is a Korean information

## Abstract

The harmful effects of alcohol consumption by adolescents have been increasingly emphasized. Thus, it is necessary to identify individual and environmental factors that encourage drinking. This study investigated factors associated with the sustainable use of alcohol (SUA) in adolescents who consume alcohol, and the possibility of future drinking (PFD) in non-drinking adolescents. Data from "The Adolescents Awareness Survey of Alcohol Encouraging Environment" by the Ministry of Health and Welfare and the Korea Health Promotion Institute (2017) were used. The survey was completed by 1,038 participant, selected through a proportional allocation extraction method, who were aged 13–18 years and lived in five cities with a population of over 1 million. The factors associated with SUA included gender ($\beta = 0.634$, $p = 0.004$), grade (8th $\beta = 1.591$, $p<0.001$, 9th $\beta = 1.674$, $p<0.001$, 10th $\beta = 1.497$, $p = 0.001$, 11th $\beta = 1.041$, $p = 0.004$, 12th $\beta = 2.610$, $p<0.001$), drinking alone ($\beta = -2.147$, $p = 0.002$), liquor commercial ($\beta = 1.644$, $p<0.001$), ease of alcohol purchase ($\beta = 1.541$, $p = 0.025$), parent's recommendation for drinking ($\beta = 1.084$, $p<0.001$), not knowing the mother's education level ($\beta = -0.685$, $p = 0.045$), positive expectancy of drinking ($\beta = 0.141$, $p<0.001$), number of pubs ($\beta = 0.303$, $p = 0.002$), internet game cafes ($\beta = 0.456$, $p = 0.019$), and karaokes ($\beta = -0.098$, $p = 0.023$) in the community. The factors associated with the PFD in non-drinkers were grade (8th $\beta = 0.531$, $p = 0.024$, 10th $\beta = 0.717$, $p = 0.035$, 12th $\beta = 1.882$, $p = 0.001$), liquor commercial ($\beta = -1.355$, $p<0.001$), parent's recommendation for drinking ($\beta = 0.783$, $p = 0.020$), positive expectancy of drinking ($\beta = 0.139$, $p<0.001$), and relationship with the father ($\beta = 0.072$, $p = 0.033$). Multidimensional interventions, including those by individuals, parents, peers, and local communities, are needed to prevent SUA and the PFD in adolescents.

## 1. Introduction

Alcohol is one of the most prevalent substances used by adolescents, and adolescent drinking has been emerging as a serious public health problem worldwide [1, 2]. Adolescents who started drinking before the age of 15 years showed a higher risk of alcohol dependence and

system site (https://www.open.go.kr/), and anyone who needs data can request data through information disclosure. Others would be able to access these data in the same manner as the authors. The authors did not have any special access privileges that others would not have. If necessary, people can join the site as a member of https://www.open.go.kr/com/login/memberLogin. do) and request variables and questionnaires (Request for environmental survey data that stimulates adolescents' drinking in 2017). Data can be provided within 10 working days upon request.

**Funding:** The author(s) received no specific funding for this work.

**Competing interests:** The authors have declared that no competing interests exist.

abuse [3], alcohol-related car accidents [4], and other unintended injuries [5] during adult-hood than those who started drinking at age 21 or older. Furthermore, since drugs such as tobacco and alcohol may act as gateways for illegal drugs, preventing alcohol consumption not only avoids or minimizes the harm caused by drinking alcohol but also delays the use of other drugs and prevents drug-related harm [6].

In previous studies, the factors that correlated to or influenced the drinking behavior of adolescents were divided into individual and environmental factors. Individual factors include socio-demographic variables such as type of school, grade level, age, gender, positive attitudes toward drinking, current smoking, knowledge about drinking and drugs, subjective stress level, depression, and positive drinking expectations that are aimed at improving sociability and sexual function [7, 8].

The factors relating to the family and friends included parental drinking or poor relation-ships with parents [8, 9], parents' recommendations of alcohol consumption [10], the number of friends who drank with, frequency of friends' drinking, and amount of alcohol that they consumed [7]. Parents' recommendation to drink in childhood displayed a particularly signifi-cant association with an increased likelihood of risky drinking later in adolescence [7].

It has been reported that non-drinkers are more likely to drink as their exposure to alcohol commercial advertising increases, as are current drinkers [11]. Liquor commercials and pro-motes to promote adolescents' positive impressions about alcohol use and encourages their intention to use alcohol. It further influences adolescents who have never drunk alcohol to have a positive attitude and intention to use alcohol through the effective messaging of adver-tising [12].

Environmental factors, including law and community ordinances, prohibit the sale and drinking of alcohol to adolescents under the age of 19 [13], however, these are not a deterrent to alcohol consumption. The higher the density of liquor stores in the community, the greater the alcohol-related harm [14]. The number of entertainment establishments and liquor stores also affects the current drinking rate of adolescents in the community [15].

As mentioned above, there are many studies on the factors that influence the current drink-ing behavior of adolescents. However, few studies have comprehensively compared individual and community environmental factors concurrently with adolescents who are not currently drinking but are likely to drink in the near future and adolescents who want to continue drink-ing without stopping in the near future.

Therefore, this study aimed to identify the influencing factors on the drinking intention of adolescents who currently drink and adolescents who do not currently drink but are likely to drink in the near future, and simultaneously compare and analyze individual and community environmental factors. Based on these results, we intended to sought evidence for reducing harmful factors that promote adolescent drinking, thereby helping to establish and implement effective adolescents drinking prevention policies.

## 2. Materials and methods

### 2.1. Participants

This descriptive cross-sectional study aimed to identify the factors influencing the SUA of drinking adolescents and the PFD of non-drinking adolescents.

The data for this study were obtained from the Environmental Drinking Survey of Adoles-cent Drinking Promotion conducted by the Ministry of Health and Welfare and the Korea Health Promotion Institute in 2017. The participants for this survey were middle and high school students nationwide, aged 13 to 18 years, and included a total of 1,045 adolescents.

Of these 1,045 adolescents, the data from 1,038 were analyzed, excluding 1) two non-responders to the question on problematic drinking, 2) one non-responder to the question on the intention to use alcohol, 3) one non-responder to the question on "desire to drink after being exposed to liquor commercial"; 4) two non-responders to the question on "people I drank with lately", and 5) one non-responder to the question on ease of alcohol purchase.

## 2.2. Instruments

**2.2.1. Dependent variables.** The dependent variable was the intention to use alcohol, which indicated the intention to use alcohol in the near future, and was measured using a tool developed by Williams, Toomey, McGovern, Wagenaar, and Perry (1995) [16]. This tool was developed primarily for self-reporting adolescent alcohol use and abuse. The sub-domains of the tool were a total of 44 items: occasion 3 items, intention to drink 4 items, marijuana use frequency 3 items (1–7 Likert scale), inhalant use frequency 3 items (1–7 Likert scale), cocaine use frequency 3 items in the alcohol use tendency domain (1–7 Likert scale), peer influence 17 items (1–5 Likert scale), self-efficacy 5 items (1–5 Likert scale), and 6 items (1–5 Likert scale). Among them, four items on the intention to use alcohol were revised to 19 years old (the original tool targeted those aged 21), the age of prohibition of drinking among adolescents in Korea. We presented four situations: (1) when the age of 19 years (adult) is reached; (2) if someone recommends alcohol within a year; (3) if someone recommends alcohol within a month; and (4) if someone recommends alcohol within a week. The reliability coefficient of alcohol use tendency, including the intention to use alcohol, of this tool was .89-.93 for boys and .85 to .92 for girls. The higher the score, the greater the intention to use alcohol. The Cronbach's α in this study was .81.

The higher the score of a drinker answering the question regarding the intention to use alcohol, the higher the SUA. The higher the score of non-drinkers responding to the question regarding the intention to use alcohol, the higher their PFD in the near future.

**2.2.2. Independent variables.** The following variables were selected by referring to the variables that affect adolescent drinking behavior in previous studies.

The positive expectancy of drinking was measured using eight items (See Fig 1). The tool consists of eight items, reconstructed by Yoon [17] from the youth types (AEQ-A) of drinking-level surveys developed by Christiansen, Smith, Roehling, and Goldman [18]. These were measured on a 5-point Likert scale, and the higher the score, the higher the positive expectancy of drinking. Cronbach's α for this study was .60.

The close relationship with parents was assessed by asking eight items (See Fig 1) and measured using a 4-point scale (from "never" to "very much"). The items are 'My parents enjoy spending time with me', 'My parents listen to me very well', 'My parents know who my friends are', 'My parents try to join me if there is anything interesting', 'My parents know where I am after school', 'My parents know where I am at night', 'My parents know what I spend on my allowance', 'My parents know what I do when I have free time'. The higher the score, the better the close relationship with parents. Cronbach's α in this study was .94.

The drinking variables were related to the parents, peers, and school (See Fig 1). These included people who drank alcohol among close friends, parents' recommendation to drink, close relationship with parents, education level of parents, and the school's disciplinary action for drinking. The school's disciplinary action measured the perceptions of drinking regulations, including punishment for students who drank alcohol at schools and drinking experiences during school events such as travel and clubs (two questions). The motivation for drinking as promoted by the media was measured by "whether you ever wanted to drink after watching a liquor commercial, an image of alcohol, or a drinking scene" (a single question).

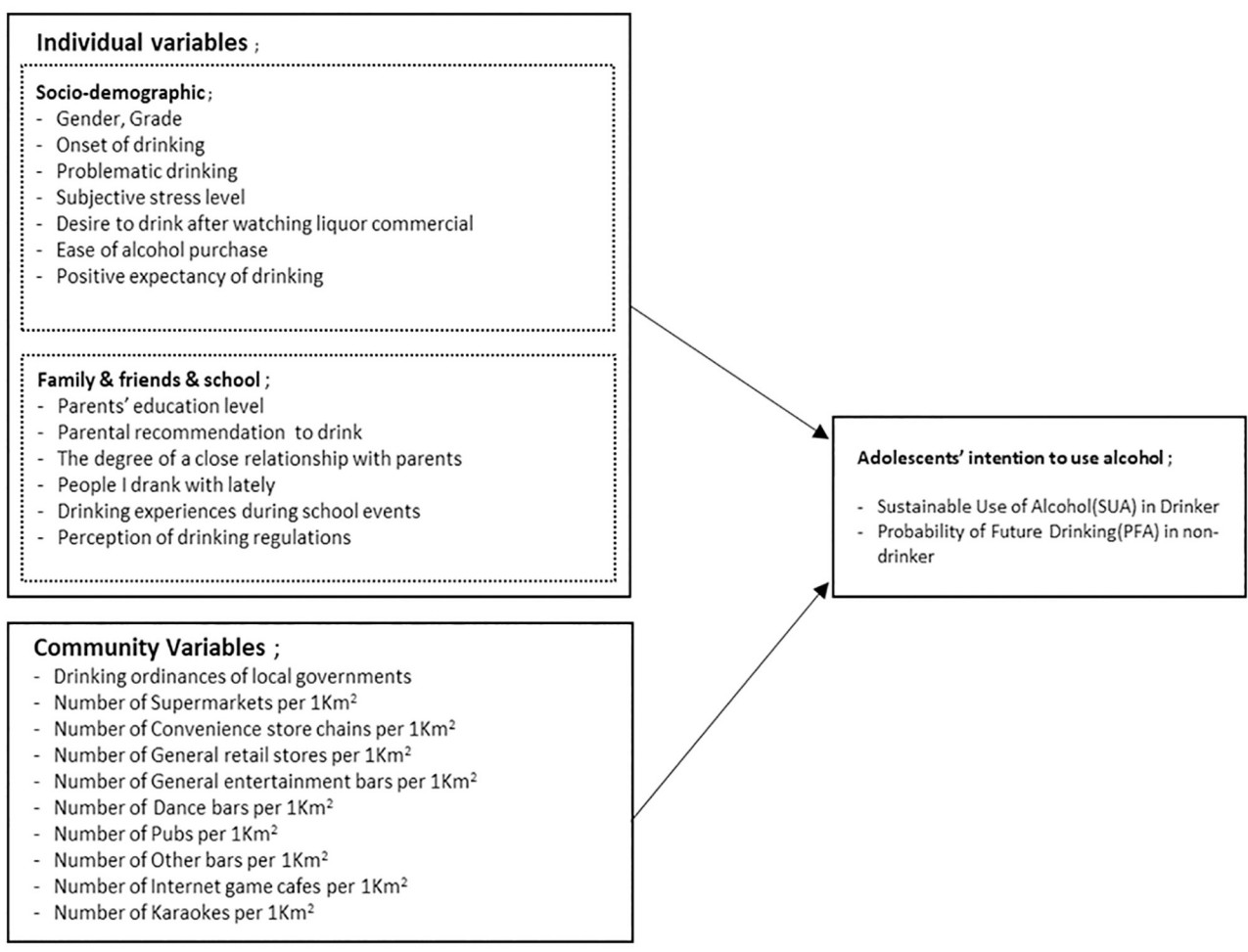

**Fig 1. Framework of this study.**

Community environmental variables were whether the local governments had drinking ordinances in 2017 by searching the municipal regulations information system [17]. The data were derived from the statistics of businesses related to alcohol consumption by the administrative district units; These administrative districts are the locations of schools where participants attend. The number of supermarkets (Korean Standard Industrial Classification Code: 47121), convenience store chains (47122), general retail (47129), general entertainment bars (56211), dance bars (56212), pubs (56213), other bars (56219), internet game cafes (91222), and karaokes (91223) were analyzed.

Variables (See Fig 1) such as gender, grade, the onset of drinking, ease of alcohol purchase, problematic drinking, subjective stress level, and parents' education level were based on the Korea Youth Risk Behavior (KYRBS) Web-based Survey (http://yhs.cdc.go.kr). The KYRBS was established in 2005 by the Centers for Disease Control and Prevention in Korea. The KYRBS is an ongoing national cross-sectional survey that assesses health-risk behaviors among middle- and high-school students and monitors progress toward achieving the national health objectives of Korea's National Health Plan 2020 and 2030. The framework of this study is shown in Fig 1.

## 2.3. Procedures

This survey was conducted from October 30, 2017, to December 14, 2017, targeting male and female adolescents aged 13–18 years who live in cities with a population of more than 1 million people. Based on the results of the 2016 Basic Education Statistics Survey, adolescents were selected according to the stratified system extraction method. The process for selecting the research participants was as follows: first, to select the schools to be surveyed, phylogenetic extraction was conducted in consideration of co-educational status and age for a balanced survey according to school type (middle school, general high school, special purpose high school, specialized high school, autonomous high school) and gender. In the selected schools, a survey was conducted targeting male and female students from classes with more than 30 students. Through this process, 1,045 students from 30 schools were surveyed.

Considering that the participants of the survey were adolescents, a professional interviewer conducted a face-to-face survey under the guidance of the school teacher only after obtaining consent from adolescents and their parents. The community data were obtained from the "Census of Establishment" conducted by Statistics Korea, which investigated all businesses with one or more employees.

## 2.4. Data analysis

All statistical analyses were carried out using SAS software (version 9.4), and results with $p <$ .05 were interpreted as being statistically significant. Variables influencing SUA in drinking adolescents and PFD in non-drinking adolescents were classified into individual variables, family and peer variables, and environmental variables.

First, the chi-square test, t-test, and ANOVA tests were performed depending on the variable to verify the differences between alcohol drinking and non-drinking adolescents. A descriptive analysis of the community circumstances variables was performed.

The variables included were: (a) individual (gender, grade, subjective stress level, onset of drinking, problematic drinking, intention to use alcohol, positive expectancy of drinking, ease of alcohol purchase, desire to drink after watching liquor commercial, etc.), (b) peers (people I drank with lately), parents (close relationship with parents, parental education level, parental recommendation to drink, etc.), school (drinking experience at school events, recognized school drinking regulations for their experience), (d) community (whether or not there was a local drinking ordinance, and the number of bars including liquor stores etc.).

Linear regression was performed to explore variables affecting adolescents' intention to use alcohol to explore the relationship between the variables, and the dependent variables were intended to use alcohol, with continuous variables.

Finally, hierarchical multiple regression was performed considering individual, parents, peer, school, and local community environmental variables simultaneously. This analysis method was used to separate individual and community environmental factors and identify their relative impacts. This means that the relationship between the variables continues even if the analysis variables differ at the individual and community levels (the number of supermarkets, etc.) [19].

## 2.5. Ethical consideration

The questionnaires and agreements developed for the survey of the youth drinking-control environment were approved by the Bioethics Committee of the Korea Health Promotion Institute (IRB 1709-HR-010-01). At the time, the investigation was conducted with the written informed consent of adolescents and parents, and the investigation was conducted with the guidance of the school teacher only if the consent of adolescents and parents was obtained.

The research data was provided with a five-digit unique number given by the investigative agency, excluding the name of the person being investigated and the school name.

## 3. Results

### 3.1. Characteristics of participants

A comparative analysis of drinking behavior among drinking and non-drinking adolescents is shown in Table 1.

There was a significant difference in alcohol consumption between 628 drinkers (60.50%) and 410 non-drinkers (39.50%). There were 604 male (63.38%) and 434 female (36.62%) students. By grade, 238 students were in 10th grade, 195 in 11th grade, 183 in 8th grade, and 162 in 7th grade.

The onset of drinking: There were 257 (40.92%) adolescents who started drinking before entering middle school (7th grade) and more than 13.0% had started by the 7th and 8th grades ($p < 0.001$).

Problematic drinking: Among the 628 drinkers, 102 (16.24%) adolescents reported problematic drinking.

Subjective stress level: 184 (29.30%) of 628 drinkers reported a high subjective stress level, as did 186 (45.37%) non-drinkers. A middle subjective stress level was reported by 262 drinkers (41.72%) and 146 non-drinkers (35.61%) ($p < 0.001$).

Desire to drink after watching liquor commercial: the idea of wanting to drink alcohol after encountering a liquor commercial was found in 393 drinkers (62.58%) and 333 non-drinkers (81.22%) ($p < 0.001$).

Ease of alcohol purchase: 134 (21.34%) of 628 drinkers reported that it was possible to buy alcohol, and 3 (0.73%) of 410 non-drinkers. In the case of alcohol purchases "being impossible," 29 (4.62%) drinkers and 10 (2.44%) non-drinkers ($p < 0.001$) believed this to be the case.

Positive expectancy of drinking: The mean score for the positive expectancy of the effect from drinking was 22.05 (±7.47) points for drinkers and 17.84 (±8.24) for non-drinkers, showing higher expectancy of a positive drinking effect in the drinking group ($p < 0.001$). When asked about sustainable drinking or intention to use alcohol, drinkers (mean score 11.56 [±3.91] and non-drinkers [mean score 8.31 [±3.04]) showed significant differences ($p < 0.001$).

### 3.2. Environment that encourages drinking

The drinking ordinance of local governments, the number of supermarkets, convenience store chains, general retail stores, general entertainment bars, dance bars, other bars, pubs, internet game cafes, and karaoke in 35 Gu (local) of five cities in Korea are shown in Table 2.

The drinking ordinances of local governments include prohibitions on drinking in public places such as parks at any age; while 20 local governments have drinking-related ordinances, 15 local governments do not. Among the number of places selling alcohol, there were 15.44 (±12.28) other bars per square kilometer followed by general retail stores, convenience store chains, and karaokes. Adolescents' intention to use alcohol increased with the number of pubs ($p = 0.033$), other bars ($p = 0.003$), internet game cafes ($p < 0.001$), and karaokes ($p < 0.001$) in the local community.

### 3.3. Factors influencing SUA in drinking adolescents

The significant variables affecting SUA for drinkers were gender, grade, people who I drank with lately, desire to drink after being exposed to liquor commercials, parents' recommendations

**Table 1. Characteristics and differences between drinkers and non-drinkers in adolescents.** (N = 1038).

| Variables | Categories | Total | Drinkers | Non-drinkers | $X^{2/t}(p)$ |
|---|---|---|---|---|---|
| | | | n (%) or Mean ± SD | | |
| Gender | Male | 604 | 398 (63.38) | 206 (50.24) | 17.582 |
| | Female | 434 | 230 (36.62) | 204 (49.76) | (<0.001) |
| Grade | 7th grade | 162 | 76 (12.1) | 86 (20.98) | 64.213 |
| | 8th grade | 183 | 91 (14.49) | 92 (22.44) | (<0.001) |
| | 9th grade | 145 | 67 (10.67) | 78 (19.02) | |
| | 10th grade | 238 | 173 (27.55) | 65 (15.85) | |
| | 11th grade | 195 | 142 (22.61) | 53 (12.93) | |
| | 12th grade | 115 | 79 (12.58) | 36 (8.78) | |
| Onset of Drinking | No drink | 410 | 0 (0) | 410 (100) | 1038 |
| | Before entering middle school | 257 | 257 (40.92) | 0 (0) | (<0.001) |
| | 7th grade | 89 | 89 (14.17) | 0 (0) | |
| | 8th grade | 84 | 84 (13.38) | 0 (0) | |
| | 9th grade | 85 | 85 (13.54) | 0 (0) | |
| | 10th grade | 85 | 85 (13.54) | 0 (0) | |
| | 11th grade | 17 | 17 (2.71) | 0 (0) | |
| | 12th grade | 11 | 11 (1.75) | 0 (0) | |
| People I drank with lately | No drink | 410 | 0 (0) | 410 (100) | 1038 |
| | Friend or classmate | 228 | 228 (36.31) | 0 (0) | (<0.001) |
| | Senior or junior | 15 | 15 (2.39) | 0 (0) | |
| | Family or Relative | 355 | 355 (56.53) | 0 (0) | |
| | alone | 25 | 25 (3.98) | 0 (0) | |
| | Etc | 5 | 5 (0.80) | 0 (0) | |
| Problematic drinking | Yes | 102 | 102 (54.26) | 0 (0) | - |
| | No | 86 | 86 (45.74) | 0 (0) | |
| Subjective stress level | None | 260 | 182 (28.98) | 78 (19.02) | 30.136 |
| | Middle | 408 | 262 (41.72) | 146 (35.61) | (<0.001) |
| | High | 370 | 184 (29.30) | 186 (45.37) | |
| Desire to drink after watching liquor commercial | Not having | 131 | 113 (17.99) | 18 (4.39) | 52.303 |
| | Middle | 181 | 122 (19.43) | 59 (14.39) | (<0.001) |
| | Having | 726 | 393 (62.58) | 333 (81.22) | |
| Ease of alcohol purchase | Impossible | 39 | 29 (4.62) | 10 (2.44) | 116.186 |
| | Possible | 137 | 134 (21.34) | 3 (0.73) | (<0.001) |
| | Not know / no response | 24 | 3 (0.48) | 21 (5.12) | |
| | Never try | 838 | 462 (73.57) | 376 (91.71) | |
| Parental recommendation to drink | No | 622 | 305 (48.57) | 317 (77.32) | 98.188 |
| | Yes | 346 | 282 (44.9) | 64 (15.61) | (<0.001) |
| | Not Applicable | 70 | 41 (6.53) | 29 (7.07) | |
| Father's level of education | ≤Middle school | 19 | 12 (1.91) | 7 (1.71) | 8.682 |
| | High school | 255 | 169 (26.91) | 86 (20.98) | (0.070) |
| | ≥ College or university | 483 | 290 (46.18) | 193 (47.07) | |
| | Absence | 32 | 22 (3.50) | 10 (2.44) | |
| | Not know | 249 | 135 (21.50) | 114 (27.8) | |

(*Continued*)

**Table 1.** (Continued)

| Variables | Categories | Total | Drinkers | Non-drinkers | $X^{2/t}(p)$ |
|---|---|---|---|---|---|
| | | | n (%) or Mean ± SD | | |
| Mother's level of education | ≤Middle school | 20 | 13 (2.07) | 7 (1.71) | 22.032 |
| | High school | 309 | 218 (34.71) | 91 (22.2) | (<0.001) |
| | ≥ College or university | 436 | 254 (40.45) | 182 (44.39) | |
| | Absence | 8 | 5 (0.80) | 3 (0.73) | |
| | Not knowing | 265 | 138 (21.97) | 127 (30.98) | |
| Drinking experience during school events | No | 936 | 526 (83.76) | 410 (100) | 65.632 |
| | Yes | 102 | 102 (16.24) | 0 (0) | (<0.001) |
| Perception of drinking regulations | No | 266 | 171 (27.23) | 95 (23.17) | 2.144 |
| | Yes | 772 | 457 (72.77) | 315 (76.83) | (0.143) |
| The degree of close relationship with father | | 1038 | 25.92 ± 5.14 | 26.21 ± 5.46 | -0.82 (0.414) |
| The degree of close relationship with mother | | 1038 | 27.89 ± 4.57 | 28.2 ± 4.73 | -1.03 (0.305) |
| Positive expectancy of drinking | | 1038 | 22.05 ± 7.47 | 17.84 ± 8.24 | 8.35 (<0.001) |
| Intention to use alcohol | | 1038 | 11.56 ± 3.91 | 8.31 ± 3.04 | 15.02 (<0.001) |

for drinking, mother's level of education, and positive expectancy of drinking. The power of this explanation is 38.0% (Table 3).

### 3.4. Factors influencing PFD in non-drinking adolescents

The variables that affect PFD of non-drinking adolescents were grade, desire to drink after watching liquor commercials, parents' recommendation for drinking, the degree of close relationship with fathers, and positive expectancy of drinking. The explanatory power was 30.3% (Table 3).

### 3.5. Factors associated with SUA by hierarchical multiple regression

The variables for the drinking adolescent's SUA were statistically significant with respect to gender, grade, people who I drank with lately, desire to drink after watching liquor commercial, ease of alcohol purchase, parental recommendation to drink, mother's level of education, positive expectancy of drinking, internet game cafes, pubs, and karaokes (Table 4).

**Table 2. Community variables.**

| Variables | Categories | n(%) or Mean ± SD | Min | Max | Mean±SD/r | p |
|---|---|---|---|---|---|---|
| Drinking ordinances of local governments | No | 15 (57.1) | - | - | 10.48±3.92 | 0.052 |
| | Yes | 20 (42.9) | | | 10.00±3.93 | |
| Supermarkets (per 1Km$^2$) | | 1.75 ± 0.89 | 0.48 | 3.40 | 0.059 | 0.055 |
| Convenience store chains (per 1Km$^2$) | | 7.82 ± 7.06 | 1.51 | 33.73 | 0.044 | 0.154 |
| General retail stores (per 1Km$^2$) | | 8.73 ± 9.59 | 1.50 | 54.92 | -0.059 | 0.059 |
| General entertainment bars (per 1Km$^2$) | | 4.39 ± 3.36 | 0.72 | 18.07 | -0.009 | 0.770 |
| Dance bars (per 1Km$^2$) | | 0.26 ± 0.32 | 0.02 | 1.42 | 0.040 | 0.193 |
| Pubs (per 1Km$^2$) | | 1.93 ± 2.04 | 0.24 | 9.14 | 0.066 | 0.033 |
| Other bars (per 1Km$^2$) | | 15.44 ± 12.28 | 2.36 | 45.7 | 0.093 | 0.003 |
| Internet game cafes (per 1Km$^2$) | | 2.14 ± 1.39 | 0.40 | 6.15 | 0.155 | <0.001 |
| Karaokes (per 1Km$^2$) | | 7.79 ± 5.88 | 1.37 | 22.98 | 0.125 | <0.001 |

Table 3. Factors influencing sustainable use of alcohol and possibility of future drinking in adolescents.

**Sustainable Use of Alcohol in Drinking Adolescents**

| Variable | Ref | Categories | Parameter Estimate | CI Lower | CI Upper | t(p) |
|---|---|---|---|---|---|---|
| Intercept | | | 7.392 | 5.495 | 9.288 | 7.66 (<0.001) |
| Gender | Female | Male | 0.683 | 0.151 | 1.214 | 2.52 (0.012) |
| Grade | 7th grade | 8th grade | 0.659 | -0.317 | 1.636 | 1.33 (0.185) |
| | | 9th grade | 0.750 | -0.309 | 1.810 | 1.39 (0.165) |
| | | 10th grade | 0.688 | -0.238 | 1.614 | 1.46 (0.145) |
| | | 11th grade | 0.870 | -0.092 | 1.832 | 1.78 (0.076) |
| | | 12th grade | 2.011 | 0.941 | 3.082 | 3.69 (<0.001) |
| People I drank with lately | Friends or classmates | Senior or junior | -0.641 | -2.325 | 1.043 | -0.75 (0.455) |
| | | Family or relative | -0.617 | -1.265 | 0.031 | -1.87 (0.062) |
| | | Alone | -2.125 | -3.498 | -0.752 | -3.04 (0.003) |
| | | Etc. | -0.526 | -3.336 | 2.284 | -0.37 (0.713) |
| Desire to drink after watching liquor commercial | Middle | Not having | -0.611 | -1.287 | 0.066 | -1.77 (0.077) |
| | | Having | 1.765 | 0.941 | 2.589 | 4.21 (<0.001) |
| Ease of alcohol purchase | Impossible | Easy | 1.264 | -0.064 | 2.592 | 1.87 (0.062) |
| | | Not knowing or Not response | -0.957 | -4.731 | 2.817 | -0.50 (0.619) |
| | | No attempt | -0.009 | -1.204 | 1.186 | -0.01 (0.988) |
| Parent's recommendation for drinking | No | Yes | 1.172 | 0.643 | 1.701 | 4.35 (<0.001) |
| | | Not Applicable | 0.684 | -0.365 | 1.734 | 1.28 (0.201) |
| Mother's level of education | ≥ Graduating from junior college | Absence | -1.757 | -4.623 | 1.109 | -1.20 (0.229) |
| | | ≤Middle school | -1.018 | -2.806 | 0.770 | -1.12 (0.264) |
| | | ≤High school | -0.286 | -0.870 | 0.298 | -0.96 (0.337) |
| | | Not knowing | -0.711 | -1.377 | -0.044 | -2.09 (0.037) |
| Drinking experience during school events | No | Yes | -0.319 | -1.046 | 0.409 | -0.86 (0.390) |

**Possibility of Future Drinking in Non-drinking Adolescents**

| Variable | Ref | Categories | Parameter Estimate | CI Lower | CI Upper | t(p) |
|---|---|---|---|---|---|---|
| Intercept | | | 6.949 | 4.797 | 9.102 | 6.35 (<0.001) |
| Grade | 7th grade | 8th grade | 0.457 | -0.364 | 1.277 | 1.09 (0.274) |
| | | 9th grade | 0.315 | -0.532 | 1.162 | 0.73 (0.465) |
| | | 10th grade | 0.637 | -0.245 | 1.518 | 1.42 (0.156) |
| | | 11th grade | 0.738 | -0.303 | 1.778 | 1.39 (0.164) |
| | | 12th grade | 1.900 | 0.793 | 3.007 | 3.38 (0.001) |
| Subjective stress level | Middle | Much | 0.759 | -0.008 | 1.527 | 1.95 (0.052) |
| | | little | 0.033 | -0.566 | 0.632 | 0.11 (0.914) |
| Desire to drink after watching liquor commercial | Middle | Not having | -1.337 | -2.154 | -0.521 | -3.22 (0.001) |
| | | Having | -1.477 | -2.919 | -0.034 | -2.01 (0.045) |
| Parent's recommendation to drink | No | Yes | 0.815 | 0.073 | 1.557 | 2.16 (0.031) |
| | | Not Applicable | -0.318 | -1.486 | 0.849 | -0.54 (0.592) |
| The degree of close relationship with the father | | | 0.139 | 0.103 | 0.175 | 7.64 (<0.001) |
| The degree of close relationship with the mother | | | 0.073 | -0.004 | 0.150 | 1.85 (0.065) |

(Continued)

**Table 3.** (Continued)

| Sustainable Use of in Drinking Adolescents | | | | | | |
| --- | --- | --- | --- | --- | --- | --- |
| Variable | Ref | Categories | Parameter Estimate | Confidence Interval | | t(p) |
| | | | | Lower | Upper | |
| Positive expectancy of drinking | | | 0.143 | 0.104 | 0.183 | 7.06 (<0.001) |
| $R^2$ = .380 | | | | | | |

| Possibility of Future Drinking in Non-drinking Adolescents | | | | | | |
| --- | --- | --- | --- | --- | --- | --- |
| Variable | Ref | Categories | Parameter Estimate | Confidence Interval | | t(p) |
| | | | | Lower | Upper | |
| Positive expectancy of drinking | | | -0.094 | -0.181 | -0.007 | -2.12 (0.035) |
| $R^2$ = .303 | | | | | | |

**Table 4. Factors influencing sustainable use of alcohol in drinking and non-drinking adolescents by hierarchical multiple regression.**

| Drinking Adolescents | | | | | | | Non-drinking Adolescents | | | | | | |
|---|---|---|---|---|---|---|---|---|---|---|---|---|---|
| Variables | Ref | Categories | Estimate | Confidence Interval | | t(p) | Variables | Ref | Categories | Estimate | Confidence Interval | | t(p) |
| | | | | Lower | Upper | | | | | | Lower | Upper | |
| Intercept | | | 6.042 | 4.365 | 7.719 | 7.40 (<0.001) | Intercept | | | 6.553 | 3.353 | 9.754 | 4.20 (<0.001) |
| Gender | Female | Male | 0.634 | 0.205 | 1.063 | 2.90 (0.004) | | | | | | | |
| Grade | 7th grade | 8th grade | 1.591 | 0.993 | 2.188 | 5.23 (<0.001) | Grade | 7th grade | 8th grade | 0.531 | 0.070 | 0.993 | 2.27 (0.024) |
| | | 9th grade | 1.674 | 1.050 | 2.299 | 5.26 (<0.001) | | | 9th grade | 0.438 | -0.442 | 1.317 | 0.98 (0.329) |
| | | 10th grade | 1.497 | 0.586 | 2.408 | 3.23 (0.001) | | | 10th grade | 0.717 | 0.050 | 1.384 | 2.11 (0.035) |
| | | 11th grade | 1.041 | 0.336 | 1.746 | 2.90 (.004) | | | 11th grade | 0.606 | -0.544 | 1.757 | 1.04 (0.300) |
| | | 12th grade | 2.610 | 1.557 | 3.663 | 4.87 (<0.001) | | | 12th grade | 1.882 | 0.761 | 3.003 | 3.30 (0.001) |
| People I drank with lately | Friends or classmates | Senior or junior | -0.562 | -2.346 | 1.222 | -0.62 (0.536) | Subjective stress level | Middle | Much | 0.807 | -0.073 | 1.687 | 1.80 (0.072) |
| | | Family or relative | -0.604 | -1.225 | 0.017 | -1.91 (0.057) | | | | | | | |
| | | Alone | -2.147 | -3.510 | -0.785 | -3.10 (0.002) | | | Little | 0.054 | -0.419 | 0.526 | 0.22 (0.824) |
| | | Etc. | -0.769 | -3.778 | 2.240 | -0.50 (0.616) | | | | | | | |
| Desire to drink after watching liquor commercial | Middle | Not having | -0.622 | -1.290 | 0.045 | -1.83 (0.068) | Desire to drink after watching liquor commercial | Middle | Not having | -1.355 | -2.111 | -0.599 | -3.53 (0.001) |
| | | Having | 1.644 | 0.903 | 2.385 | 4.36 (<0.001) | | | Having | -1.459 | -3.113 | 0.195 | -1.74 (0.084) |
| Ease of alcohol purchase | Impossible | Possible | 1.541 | 0.194 | 2.887 | 2.25 (0.025) | The degree of a close relationship with the father | | | 0.072 | 0.006 | 0.139 | 2.15 (0.033) |
| | | Not know/ no response | -0.377 | -2.184 | 1.431 | -0.41 (0.682) | The degree of a close relationship with the mother | | | -0.092 | -0.189 | 0.006 | -1.85 (0.065) |
| | | Not try | 0.198 | -0.903 | 1.298 | 0.35 (0.724) | | | | | | | |
| Parent's recommendation for drinking | No | Yes | 1.084 | 0.652 | 1.516 | 4.93 (<0.001) | Parent's recommendation for drinking | No | Yes | 0.783 | 0.124 | 1.443 | 2.34 (0.020) |
| | | Not Applicable | 0.677 | -0.323 | 1.677 | 1.33 (0.184) | | | Not Applicable | -0.403 | -1.268 | 0.462 | -0.92 (0.360) |
| Mother's level of education | ≥ College or university | Absence | -1.994 | -4.122 | 0.133 | -1.84 (0.066) | | | | | | | |
| | | ≤Middle school | -1.111 | -2.281 | 0.059 | -1.87 (0.063) | | | | | | | |
| | | High school | -0.370 | -0.930 | 0.189 | -1.30 (0.194) | | | | | | | |
| | | Not knowing | -0.685 | -1.355 | -0.015 | -2.01 (0.045) | | | | | | | |
| Drinking experience during school events | No | Yes | -0.264 | -0.939 | 0.411 | -0.77 (0.442) | | | | | | | |

(*Continued*)

**Table 4.** (Continued)

| Drinking Adolescents | | | | | | | Non-drinking Adolescents | | | | | | |
|---|---|---|---|---|---|---|---|---|---|---|---|---|---|
| **Variables** | **Ref** | **Categories** | **Estimate** | **Confidence Interval** | | **t(p)** | **Variables** | **Ref** | **Categories** | **Estimate** | **Confidence Interval** | | **t(p)** |
| | | | | **Lower** | **Upper** | | | | | | **Lower** | **Upper** | |
| Positive expectancy of drinking | | | 0.141 | 0.103 | 0.180 | 7.17 (<0.001) | Positive expectancy of drinking | | | 0.139 | 0.094 | 0.183 | 6.17 (<0.001) |
| Internet game cafes (per 1Km$^2$) | | | 0.456 | 0.075 | 0.838 | 2.35 (0.019) | Internet game cafes (per 1Km$^2$) | | | 0.155 | -0.098 | 0.409 | 1.20 (0.229) |
| Pubs (per 1Km$^2$) | | | 0.303 | 0.110 | 0.496 | 3.08 (0.002) | | | | | | | |
| Other bars (per 1Km$^2$) | | | -0.010 | -0.037 | 0.016 | -0.78 (0.434) | | | | | | | |
| Karaokes (per 1Km$^2$) | | | -0.098 | -0.182 | -0.014 | -2.29 (0.023) | | | | | | | |
| $R^2$ = .395, ICC = .13 | | | | | | | $R^2$ = .314, ICC = .12 | | | | | | |

SUA of male drinkers was 0.634 points higher than that of female drinkers ($p$ = 0.004), and this increased as the grade went up. Compared to friends or classmates I drank with lately, drinking alone was 2.147 points lower ($p$ = 0.002). The number of people who desire to drink after watching a liquor commercial increased by 1.644 points compared to "Middle" ($p$<0.001). SUA was 1.541 points higher among those who experienced it as "ease of alcohol purchase" compared to those who experienced it as "impossible to buy alcohol" ($p$ = 0.025). Drinking increased by 1.084 points when adolescents recommended by parents ($p$<0.001) compared to when parents did not recommend that their children drink alcohol. Compared to cases in which the mother's educational level was higher than that of a college graduate, SUA of adolescents who responded "do not know" decreased by 0.685 points ($p$ = 0.045). For drinkers, SUA increased by 0.141 points when positive expectancy of drinking increased by 1 ($p$<0.001). However, the rate of increase was not higher than that of non-drinking adolescents, which is believed to be due to the fact that they have already experienced the reality of expectancy of drinking. In the local community, when the number of pubs, increased by 1 per unit area of 1 km$^2$, SUA increased by 0.303 points ($p$ = 0.002), and when the number of internet game cafés increased by 1 per unit area of 1 km$^2$, SUA increased by 0.456 ($p$ = 0.019). On the other hand, SUA decreased by 0.098 points when the number of karaokes increased by 1 per unit area of 1 km$^2$ ($p$ = 0.023).

### 3.6. Factors associated with PFD by hierarchical multiple regression

The variables associated with PFD of non-drinkers were grade, desire to drink after watching a liquor commercial, parents' recommendation for drinking, the close relationship with their father, and positive expectancy of drinking (Table 4).

PFD increased in the 8th grade (β = 0.531, $p$ = 0.024), 10th grade (β = 0.717, $p$ = 0.035), and 12th grade (β = 1.882, $p$ = 0.001), when compared to the 7th grade among non-drinkers. This is believed to be because of 12th graders' (equivalent to high school seniors) subjective stress level during college entrance exams, and the students approaching the age of 19, which is a legally allowed drinking age in Korea. PFD was reduced by 1.355 points when adolescents responded to "Not having" compared to "Middle" in a desire to drink after watching a liquor commercial ($p$ = 0.001). PFD increased by 0.783 points when adolescents responded to "Yes" of parental recommendations for drinking compared to "No" ($p$ = 0.020). PFD increased by

0.139 points, as positive expectancy of drinking increased by 1 ($p<0.001$). As their relationship with their fathers increased by 1, the PFD of adolescents increased by 0.072 points ($p = 0.033$). On the other hand, the number of internet game cafes per 1 $km^2$ did not significantly affect the PFD of non-drinkers.

## 4. Discussion

This study identified the sustainable use of alcohol (SUA) in drinking adolescents and the possibility of future drinking (PFD) in non-drinking adolescents, living in large cities, and analyzed the factors affecting SUA and PFD using the adolescents' individual and community environmental variables. The analysis was performed using hierarchical multiple regression. The results of the study show that the factors affecting SUA in drinking adolescents are gender, grade, people who I drank with lately, desire to drink after watching liquor commercials, ease of alcohol purchase, parents' recommendation for drinking, education level of the mother, positive expectancy of drinking at the individual level. These factors were found to influence the SUA. The associated community factors were the number of pubs, internet game cafés, and karaokes in the local community. The power of SUA explanation was 39.5%.

The factors influencing PFD in non-drinking adolescents was grade, desire to drink after watching liquor commercials, parents' recommendation for drinking, positive expectancy of drinking, and the close relationship with their father. The example of not thinking about desire to drink after watching a liquor commercial was found to have a negative effect on PFD. There were no statistically significant variables for PFD associated with local community factors. The power of PFD explanation was 31.4%.

In this study, parents' recommendations for drinking were found to increase drinking intentions among both drinking and non-drinking adolescents, which is consistent with the results of previous studies that reported that the more parents drink, the more their children experience drinking alcohol [20]. In Korea, the experience of being encouraged to drink increases during family events [21]. This might be due to the social perception that adolescents can learn proper drinking etiquette from adults. It shows that a tolerant social atmosphere toward alcohol and parents' recommendations for drinking encouraged adolescents to drink, since it is interpreted as tacit acceptance of drinking, thus alleviating the guilt about ignoring the ban on drinking under the age of 19. This suggests that the culture of tolerance to drinking is deeply embedded in each family in Korea.

The parents' role is most important in preventing adolescent's drinking, and parental restrictions are necessary because strict parental control can lower the adolescent drinking rate [18]. Since adolescence is a period in which attitudes toward drinking are formed, it is necessary to change the false belief that parents' recommendations for drinking will have a positive effect on improving the relationship with children or on the establishment of good drinking attitudes through accurate education on drinking [22].

Among parent-child relationships, the relationship with the father was not significantly associated with alcohol consumption in drinking adolescents but was only found to be a weak influencing variable for PFD in non-drinking adolescents. This differs from the results of a previous study that reduced levels of alcohol use/misuse, delayed the onset of drinking, and such intentions when the parent-child relationship was good [23].

Liquor commercials appeared to be associated with the intention to use alcohol in both drinking and non-drinking adolescents. Drinking adolescents had increased SUA following being exposed to liquor commercials. However, the PFD for non-drinking adolescents decreased when they said they had no intention of drinking after watching liquor commercials. This result was partially consistent with another finding that the preference for alcohol

advertising was high and the likelihood of drinking increased as exposure to advertising increased, with drinkers indicating that they drink more, and non-drinkers reporting a high intention to drink [15, 24]. TV, mass media, print commercials, and internet social media, which can be easily accessed by adolescents, also targets related to alcohol consumption.

Positive emotional experiences, achievements, individuality, and camaraderie about drinking, observed through the media can influence the onset, duration, and frequency of drinking [25, 26]. Therefore, continuous monitoring and policy interventions in various media are necessary [27].

SUA increased for drinking adolescents when positive expectancy of drinking increased. Positive expectancy of drinking is a strong predictor of SUA. In the case of drinking adolescents, it was consistent with previous studies that the higher the positive expectations about drinking, the higher the sustained drinking [28]. Positive expectancy of drinking has been reported to have a significant effect on alcohol consumption [28]. On the other hand, PFD increased for non-drinking adolescents when positive alcohol expectancy increased. In previous studies, non-drinking adolescents were found to be less likely to drink in the future [21], suggesting that positive expectancy of drinking does not affect the intention to drink in non-drinking adolescents.

Among the factors related to individual variables, adolescents' intentions to use alcohol decreased when considering drinking alone compared to drinking with peers. In adolescence, imitation plays an important role in the onset of drinking, and drinking peer influences continuous drinking [29]. Usually, drinking alone reduces simple negative emotions [30, 31], and it is assumed that this will not lead to continuous drinking.

The number of pubs and internet game cafés was found to be a local community factor affecting SUA in drinking adolescents. Internet use amongst adolescents is positively related to drinking; the study is based on the evidence that exposure to alcohol through the Internet and social networking sites (SNS) increases, and the drinking rate increases through social online activities with friends [32]. The fact that there are many places for adolescents to meet and buy alcohol in the community can be interpreted as the provision of an atmosphere in the community that is tolerant of drinking. Internet use also leads to decreased cognitive and behavioral self-control, which increases problematic drinking behavior [33]. This study found that drinking environment in the community influenced adolescents' drinking intentions. In the case of drinking prevention policy, it is necessary to actively promote policies that lead to environmental control, such as the regulation of the number of pubs and places where alcohol is sold in the local community.

Current drinking prevention education for adolescents is formal and fragmentary, but there has been no increase in the prevalence of drinking among adolescents. However, in recent years, ease of alcohol purchase has been on the rise, and the experience rate of drinking prevention education has decreased significantly [34]. It will be necessary to develop a specific prevention drinking program for non-drinking adolescents, as well as active stopping drinking campaigns and individual counseling for drinking adolescents. Early drinking onset in adolescence is associated with an early drug abuse behavior that may lead to problematic drinking in adulthood, which also affects health behavior problems [35, 36]. Therefore, if there is no solution to the current drinking behavior of drinking adolescents, it may become a risk factor for future social problems. It is thus necessary to effectively deliver messages, such as the 'principle of zero tolerance' and 'prohibition of drinking under the age of 19' to legal regulations for adolescents.

## 5. Conclusions

One of the goals of the adolescent drinking prevention policy is to delay the initiation of alcohol consumption by changing the legal drinking age. This is because the earlier the drinking

age, the greater the risk of alcohol-related disability regarding adolescents' growth and development, and the higher the likelihood of substance abuse or addiction during adulthood. In this context, it is essential to prevent or reduce drinking among adolescents by identifying factors related to their drinking intentions and considering the possibility of drinking for non-drinkers who have not yet started drinking, especially adolescents who have experimented only once or twice out of curiosity.

This cross-sectional study aimed to examine the drinking behavior of adolescents. The study had some limitations; since this study had a cross-sectional design, it was not possible to prove the direction of the association. To prevent the possibility and sustainability of drinking of adolescent drinking in advance, it is important to control the influence of factors that promote drinking in the community and to prepare a specific and strong policy that can limit it.

## Author Contributions

**Conceptualization:** Eun-A Park, Ae-Ri Jung.

**Data curation:** Sungyong Choi.

**Formal analysis:** Sungyong Choi.

**Methodology:** Eun-A Park, Sungyong Choi.

**Supervision:** Eun-A Park, Ae-Ri Jung.

**Visualization:** Eun-A Park, Ae-Ri Jung.

**Writing – original draft:** Eun-A Park, Ae-Ri Jung.

**Writing – review & editing:** Eun-A Park, Ae-Ri Jung.

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
