## [Decision Letter · Decision Letter 0]

18 Nov 2021

PONE-D-21-30940Analysis of related factors for adolescents’ intention to use alcohol in KoreaPLOS ONE

Dear Dr. Jung,

Thank you for submitting your manuscript to PLOS ONE. After careful consideration, we feel that it has merit but does not fully meet PLOS ONE’s publication criteria as it currently stands. Therefore, we invite you to submit a revised version of the manuscript that addresses the points raised during the review process.

In addition to the review comments below:

It would be helpful to ensure consistency in presenting the p values (3 decimal points)Consider presenting the 95% confidence intervals.Make to detail whether the analysis was adjusted for the survey design ( sampling weights, stratification and clustering). Please submit your revised manuscript by Jan 02 2022 11:59PM. If you will need more time than this to complete your revisions, please reply to this message or contact the journal office at plosone@plos.org. Please include the following items when submitting your revised manuscript:A rebuttal letter that responds to each point raised by the academic editor and reviewer(s). You should upload this letter as a separate file labeled 'Response to Reviewers'.A marked-up copy of your manuscript that highlights changes made to the original version. You should upload this as a separate file labeled 'Revised Manuscript with Track Changes'.An unmarked version of your revised paper without tracked changes. You should upload this as a separate file labeled 'Manuscript'.

We look forward to receiving your revised manuscript.

Kind regards,

Joel Msafiri Francis, MD, MS, PhD

Academic Editor

PLOS ONE

Journal Requirements:

a) Did participants provide their written or verbal informed consent to participate in this study?

Additional Editor Comments:

In addition to the review comments:

1. It would be helpful to ensure consistency in presenting the p values (3 decimal points)

2. Consider presenting the 95% confidence intervals.

3. Make to detail whether the analysis was adjusted for the survey design ( sampling weights, stratification and clustering).

Reviewers' comments:

Reviewer's Responses to Questions

**Comments to the Author**

1. Is the manuscript technically sound, and do the data support the conclusions?

Reviewer #1: Partly

Reviewer #2: Partly

2. Has the statistical analysis been performed appropriately and rigorously? 

Reviewer #1: I Don't Know

Reviewer #2: Yes

3. Have the authors made all data underlying the findings in their manuscript fully available?

Reviewer #1: Yes

Reviewer #2: Yes

4. Is the manuscript presented in an intelligible fashion and written in standard English?

Reviewer #1: No

Reviewer #2: Yes

5. Review Comments to the Author

Reviewer #1: The manuscript would benefit from a complete grammar and spell check.

Issues related to the content are to be found in the attached reviewer's tracked changes and edits.

Some key measures reported e.g. Problem drinking and Stress have not been adequately explained.

In addition, the background needs to highlight the issue of underage drinking more strongly in order to build an argument and strengthen the justification for this study.

The manuscript would have benefited from a conceptual framework e.g. ecological model to explain the multiple influences on adolescent drinking or intention to drink.

Reviewer #2: 1. The introduction is framed in a way that indicates that this study is redundant as the introduction points out existence of the very evidence that this paper reports about. The authors will do well to highlight the gaps in research that this study addresses.

2. The content of the paper is overwhelming given the numerous variables included. The authors would do well to present a (clearer) conceptual or theoretical framework that more clearly explains the inclusion of all these variables. They already allude to it in pointing out that factors that contribute to adolescent alcohol use are layered from individual level to community level.

3. The univariate results presented under 3.3 and 3.4 (which I gather are a check for significant predictors to be included in the hierarchical regression model) do not need to be discussed in that much detail. The authors can simply indicate which variables were significant and which were not in the findings/results section, and completely leave them out of the discussion section.

4. It seems that the authors are presenting the results of table 2 as though an association was tested between the drinking venues density variables and the dependent variables. The authors need to present results of an association if indeed they hope to draw such conclusions.

5. The manuscript needs thorough editing:

a. “Previous studies on the drinking behaviors of adolescents have found that the individual factors were related to positive attitudes toward drinking, including current smoking, …” do the authors mean substance use (rather than drinking), given the next part of the sentence?

b. “On the other hand, the higher the score of nondrinkers responding to the question regarding the tendency to use alcohol, ...” do the authors mean intention rather than tendency?

c. “Positive expectations for drinking used eight positive expectations’ measures from the 16 categories of drinking scale.” This sentence is currently not so clear.

d. “This is because the earlier the drinking age, the greater the risk of disability regarding adolescents’ growth and development” – alcohol-related disability rather than any disability?

6. “Alcohol is the most prevalent substance that adolescents abuse among harmful drugs such as cigarettes, drugs, and hallucinogenic substances” – do the authors mean ‘one of’ rather than ‘the most’? It would also be useful for the authors to indicate whether these are global or national patterns.

7. “The dependent variable was the intention to use alcohol, which indicated the intention to use alcohol in the near future, and was measured using a tool developed by the University of Minnesota” More information about the tool is needed. What is its name, when was it developed, what are its psychometric properties of the tool, how are the questions phrased, how it is scored, what is the full reference of the source? The authors say they presented four situations - are these 4 selected from x, if so from how many, and why these 4?

8. “The following procedure was used to estimate the internal consistency of the scores to examine the reliability of the instruments used for data collection.” Which following procedure?

9. “There was a significant difference in alcohol consumption between 628 (60.5%) and 410 non-drinkers (39.5%).” What do the authors mean by alcohol consumption here? Also, 'drinkers' seems to be missing.

10. Table 1, rows 3 and 4 – the figures in the p-value column do not seem to be p-values (possibly sample size). I would also suggest that the Chisquare and t-test results be presented in turn rather than mixed as they are now in the table.

11. A few points in the discussion are misaligned with the findings and/or need further engagement:

a. “However, drinking and non-drinking adolescents are still required to understand drinking prevention education and that the rationale for this is to promote their health.” - This conclusion appears unrelated to the authors’ summary of the findings in this paragraph.

b. How do the authors explain the negative association between positive expectations and low likelihood of drinking among non-drinking adolescents?

c. “Among the factors related to individual circumstances, an association was found between drinking adolescents intentions to drink with friends or schoolmates compared to drinking alone.” – this is confusing as there is reference to an association as well as a comparison. I suggest the authors rephrase this and be sure to present in a way that is supported by the research question and findings.

d. “This study found that the alcohol environment in the community was directly associated with adolescents’ drinking intentions.” – as indicated earlier, there is no evidence of this association from the findings that are currently presented.

e. “In addition, since only the variables included in this study were considered, a follow-up study considering other factors would be necessary.” – this statement is quite broad, it would be useful for the authors to provide more specific detail. What do the authors think is missing in understanding drinking intentions among adolescents?

Other general comments

I wonder whether the authors may consider using the term participants as it has become a more preferred term than subjects in the research field.

“In the case of drinking adolescents, it was consistent with previous studies that the higher the positive expectations about drinking, the higher the drinking continuity.” – sustained drinking and/or future drinking? Consistency in use of terms is needed and this needs to be checked throughout the manuscript.

References are needed for these statements:

“Local governments' drinking ordinances include prohibitions on drinking in public places such as parks, while 20 local governments have drinking-related ordinances although 15 do not.” – a reference is needed for this statement.

“Positive emotional experiences, achievements, individuality, and camaraderie about drinking, observed through the media can influence the onset, duration, and frequency of drinking.”

6. PLOS authors have the option to publish the peer review history of their article (what does this mean?). If published, this will include your full peer review and any attached files.

Reviewer #1: No

Reviewer #2: No

---

## [Author Response · Author response to Decision Letter 0]

21 Jan 2022

The following are the changes made in the revised manuscript:

- Total : 

1. We revised presenting the p values (3 decimal points) and completed grammar, spell check, and terms check throughout the manuscript.

2. We revised ethics statement on 2.5. Ethical consideration.

3. We added additional information regarding the survey on parts of 2.2. instruments.

4. Data Availability : This is a Korean information disclosure system site(https://www.open.go.kr/), and anyone who needs data can request data through information disclosure. Data can be provided within 10 working days upon request.

- Introduction : We revised to improve evidence of this study

- Materials and Methods : 

1. We added more information about survey tool and variables. 

2. We have added framework of study. 

- Results : We presented the results of additional analysis of chi-square, t-test in the table1.

- Discussion : We revised on the parts of discussion based upon the reviewer’s suggestion.

---

## [Decision Letter · Decision Letter 1]

9 May 2022

PONE-D-21-30940R1Analysis of related factors for adolescents’ intention to use alcohol in KoreaPLOS ONE

Dear Dr. Jung,

Thank you for submitting your manuscript to PLOS ONE. After careful consideration, we feel that it has merit but does not fully meet PLOS ONE’s publication criteria as it currently stands. Therefore, we invite you to submit a revised version of the manuscript that addresses the points raised during the review process.

We look forward to receiving your revised manuscript.

Kind regards,

Joel Msafiri Francis, MD, MS, PhD

Academic Editor

PLOS ONE

Journal Requirements:

Reviewers' comments:

Reviewer's Responses to Questions

**Comments to the Author**

1. If the authors have adequately addressed your comments raised in a previous round of review and you feel that this manuscript is now acceptable for publication, you may indicate that here to bypass the “Comments to the Author” section, enter your conflict of interest statement in the “Confidential to Editor” section, and submit your "Accept" recommendation.

Reviewer #1: All comments have been addressed

2. Is the manuscript technically sound, and do the data support the conclusions?

Reviewer #1: Partly

3. Has the statistical analysis been performed appropriately and rigorously? 

Reviewer #1: Yes

4. Have the authors made all data underlying the findings in their manuscript fully available?

Reviewer #1: Yes

5. Is the manuscript presented in an intelligible fashion and written in standard English?

Reviewer #1: No

6. Review Comments to the Author

Reviewer #1: The authors have partly addressed comments raised, but the manuscript still requires further work, particularly on grammar, spelling and quality check. Many of the variables have also not been fully explained, and it is difficult to make a decision on the quality of the analysis without full access to what variables measured e.g. no items are provided for certain variables to give the reader a sense of what was asked. The manuscript is also written in colloquial language in some places, and general statements are made such as "a lot of stress" which does not quantify what this means in the context of the study.

7. PLOS authors have the option to publish the peer review history of their article (what does this mean?). If published, this will include your full peer review and any attached files.

Reviewer #1: No

---

## [Author Response · Author response to Decision Letter 1]

21 Jun 2022

We agree with reviewer's suggestion. So based upon the suggestion,

1) We revised on this term (gender) and keep consistent.

2) We checked and changed the grammar

3) We have added on parts of 2.2.2. independent variables.

4) We have revised “ a lot of stress” to a high level of stress and changed this mark(***-> p<0.001) on table 1.

5) We revised unclear terms on Table 3&4.

---

## [Decision Letter · Decision Letter 2]

10 Aug 2022

PONE-D-21-30940R2Analysis of related factors for adolescents’ intention to use alcohol in KoreaPLOS ONE

Dear Dr. Jung,

Thank you for submitting your manuscript to PLOS ONE. After careful consideration, we feel that it has merit but does not fully meet PLOS ONE’s publication criteria as it currently stands. Therefore, we invite you to submit a revised version of the manuscript that addresses the points raised during the review process.

We look forward to receiving your revised manuscript.

Kind regards,

Joel Msafiri Francis, MD, MS, PhD

Academic Editor

PLOS ONE

Journal Requirements:

Reviewers' comments:

Reviewer's Responses to Questions

**Comments to the Author**

1. If the authors have adequately addressed your comments raised in a previous round of review and you feel that this manuscript is now acceptable for publication, you may indicate that here to bypass the “Comments to the Author” section, enter your conflict of interest statement in the “Confidential to Editor” section, and submit your "Accept" recommendation.

Reviewer #1: All comments have been addressed

Reviewer #2: All comments have been addressed

2. Is the manuscript technically sound, and do the data support the conclusions?

Reviewer #1: Yes

Reviewer #2: Yes

3. Has the statistical analysis been performed appropriately and rigorously? 

Reviewer #1: Yes

Reviewer #2: Yes

4. Have the authors made all data underlying the findings in their manuscript fully available?

Reviewer #1: Yes

Reviewer #2: Yes

5. Is the manuscript presented in an intelligible fashion and written in standard English?

Reviewer #1: Yes

Reviewer #2: Yes

6. Review Comments to the Author

Reviewer #1: The manuscript is vastly improved after the revisions. It addresses an important, but well described topic. Hence authors should pay special attention to drawing out the new and innovative aspects of their study. There remain a list of outstanding comments and needs for clarifications (see attached reviewer's comments) for the authors to address before it is wholly satisfactory. I also recommend that the paper be run through an editing tool such as Grammarly to assist with final edits.

I wish the authors well with their final revision and submission.

Reviewer #2: The authors need to attend to some issues.

1. In the abstract, figures for the variable 'grade' are omitted.

2. It is unclear what are protective and risk factors in the way that the results are written up in the abstract. Perhaps group them as such.

2. Please decide on the number of decimal points (2 or 3) for the Cronbach's alphas and be consistent.

2. Figure 1 is shown twice

3. Check consistency of the variable: "desire to drink alcohol after watching liquor advertisement" throughout the manuscript, sometime 'desire to' is left out

7. PLOS authors have the option to publish the peer review history of their article (what does this mean?). If published, this will include your full peer review and any attached files.

Reviewer #1: **Yes: **Dr Leane Ramsoomar

Reviewer #2: No

---

## [Author Response · Author response to Decision Letter 2]

20 Aug 2022

1) We agree with comments, so we revised words and grammar. 

2) We added Ethical considerations and discussion.

---

## [Decision Letter · Decision Letter 3]

20 Sep 2022

PONE-D-21-30940R3Analysis of related factors for adolescents’ intention to use alcohol in KoreaPLOS ONE

Dear Dr. Jung,

Thank you for submitting your manuscript to PLOS ONE. After careful consideration, we feel that it has merit but does not fully meet PLOS ONE’s publication criteria as it currently stands. Therefore, we invite you to submit a revised version of the manuscript that addresses the points raised during the review process.

We look forward to receiving your revised manuscript.

Kind regards,

Joel Msafiri Francis, MD, MS, PhD

Academic Editor

PLOS ONE

Journal Requirements:

Additional Editor Comments (if provided):

Please address the additional minor comments - mainly grammatical errors.

Reviewers' comments:

Reviewer's Responses to Questions

**Comments to the Author**

1. If the authors have adequately addressed your comments raised in a previous round of review and you feel that this manuscript is now acceptable for publication, you may indicate that here to bypass the “Comments to the Author” section, enter your conflict of interest statement in the “Confidential to Editor” section, and submit your "Accept" recommendation.

Reviewer #1: All comments have been addressed

Reviewer #2: All comments have been addressed

2. Is the manuscript technically sound, and do the data support the conclusions?

Reviewer #1: Yes

Reviewer #2: Yes

3. Has the statistical analysis been performed appropriately and rigorously? 

Reviewer #1: Yes

Reviewer #2: Yes

4. Have the authors made all data underlying the findings in their manuscript fully available?

Reviewer #1: Yes

Reviewer #2: Yes

5. Is the manuscript presented in an intelligible fashion and written in standard English?

Reviewer #1: Yes

Reviewer #2: Yes

6. Review Comments to the Author

Reviewer #1: Save for the minor grammatical issues in the now reviewed manuscript, which need to be addressed in the attached comments, the manuscript can be accepted for publication.

Thank you

Reviewer #2: Authors need to please change singular 'participant' to plural 'participants' and 'liquor commercial' to 'liquor commercials'.

7. PLOS authors have the option to publish the peer review history of their article (what does this mean?). If published, this will include your full peer review and any attached files.

Reviewer #1: No

Reviewer #2: No

---

## [Author Response · Author response to Decision Letter 3]

21 Sep 2022

We agree with this suggestion. So based upon the suggestion, we revised and highlight to red.

---

## [Editor Report · Decision Letter 4]

27 Sep 2022

Analysis of related factors for adolescents’ intention to use alcohol in Korea

PONE-D-21-30940R4

Dear Dr. Jung,

We’re pleased to inform you that your manuscript has been judged scientifically suitable for publication and will be formally accepted for publication once it meets all outstanding technical requirements.

Kind regards,

Joel Msafiri Francis, MD, MS, PhD

Academic Editor

PLOS ONE
---

## [Editor Report · Acceptance letter]

30 Sep 2022

PONE-D-21-30940R4 

Analysis of related factors for adolescents’ intention to use alcohol in Korea 

Dear Dr. Jung:

I'm pleased to inform you that your manuscript has been deemed suitable for publication in PLOS ONE. Congratulations! Your manuscript is now with our production department. 

Kind regards, 

on behalf of

Dr. Joel Msafiri Francis 

Academic Editor

PLOS ONE